# The Impact of Parental Alienating Behaviours on the Mental Health of Adults Alienated in Childhood

**DOI:** 10.3390/children9040475

**Published:** 2022-03-30

**Authors:** Suzanne Verhaar, Mandy Louise Matthewson, Caitlin Bentley

**Affiliations:** 1School of Psychological Sciences, University of Tasmania, Private Bag 30, Hobart, TAS 7001, Australia; sverhaar@utas.edu.au; 2Royal Adelaide Hospital, Port Road, Adelaide, SA 5000, Australia; caitlin.reed@utas.edu.au

**Keywords:** parental alienation, alienating parent/s, targeted parent/s, alienated child/ren, mental health, intergenerational transmission, child abuse

## Abstract

This study qualitatively investigated the mental health of adults exposed to parental alienating behaviours in childhood. Research suggests that exposure to parental alienating behaviours in childhood can have a profound impact on the mental health of those children later in life, including experiencing anxiety disorders and trauma reactions. An international sample of 20 adults exposed to parental alienating behaviours in childhood participated in semi-structured interviews on their experience and its impact. Four themes were identified: mental health difficulties, including anxiety disorders and trauma reactions, emotional pain, addiction and substance use, and coping and resilience. Intergenerational transmission of parental alienation was found. Confusion in understanding their experience of alienation, the mental health sequelae, and elevated levels of suicidal ideation were found. This study demonstrated the insidious nature of parental alienation and parental alienating behaviours and provided further evidence of these behaviours as a form of emotional abuse.

## 1. Introduction

“No matter what you think of the other party, these children are one-half of each of you. Remember that, because every time you tell your child what an ‘idiot’ his father is, or what a ‘fool’ his mother is, you are telling the child half of him is bad” (From a divorce ruling, Judge Michael Haas, MN, USA, 2001).

Parental alienation has been defined by some as a serious mental condition in children exposed to parental alienating behaviours [1,2]. Parental alienation can be identified through the presence of five factors: (1) the child refuses, opposes, or avoids a relationship with a parent; (2) the child had a positive relationship with that parent before they rejected them; (3) there is no evidence of abuse or neglect perpetrated by the rejected parent; (4) the other parent has used multiple parental alienating behaviours; (5) the child exhibits behavioural manifestations of parental alienation [3]. Consideration of parental alienation as a mental condition diagnosed in children has attracted criticism in the literature [4]. Others take a broader view of parental alienation with a focus on the nature and outcome of parental alienating behaviours [5]. Parental alienating behaviours are considered a complex cluster of strategies used by alienating parents to damage and sever the relationship between the child and the child’s other parent (targeted parent). It has been suggested that parental alienating behaviours are best understood in the context of family violence, whereby parental alienation is the outcome of an abusive process perpetrated by the alienating parent [5]. Parental alienating behaviours can include the alienating parent discrediting the targeted parent by sabotaging, undermining, and manipulating their relationship with the child [6]. It is thought that at least 19% of the population in the United States has been exposed to parental alienating behaviours [7]. This contrasts with parental estrangement, where the parent–child relationship has been negatively affected, usually with a sound rationale for the child’s rejection of the parent [5,8].

### 1.1. The Alienating Parent

To understand the experience of alienated children, it is important to review the characteristics and behaviours of the alienating parent who creates the predicament for the child. It has been suggested that alienating parents present with paranoid, histrionic, or narcissistic personality traits and have affective disorders, suicidal ideation, and lack resilience around separation and loss. They also tend to have dysfunctional family histories and poor relationships with their parents. Their desire for vengeance, coupled with feelings of anger and frustration, may inhibit them from having a more moderate view of the relationship between the child and the targeted parent [5]. As a result, the alienating parent engages in behaviours and processes that prioritise their own needs over and above the child’s needs [9,10].

A study examining the experiences of 40 alienated adult children in North America found that alienating parents used similar tactics to cult leaders to alienate the child from the targeted parent [11]. The similarities included: requiring excessive devotion; the use of emotional manipulation and persuasion techniques to reinforce dependency; denigration of the targeted parent (or outside influence for cult followers); creating the impression that the targeted parent is dangerous; deceiving the child about the targeted parents’ feelings; withdrawal of love as punishment and erasing the memory of the targeted parent [11]. The idea that alienating parents use similar tactics to cult leaders is supported by Haines et al. [5], who argued that processes evident in cultic groups, such as psychologically abusive group processes, isolation, control, and indoctrination, also held true in the case of the alienating parents’ tactical agenda.

### 1.2. The Targeted Parent

Targeted parents can present with a history of passivity, emotional constriction, and over-accommodation of demands made by alienating parents [12,13]. Targeted parents may also avoid seeking to maintain a relationship with their child for fear of being rejected or hurt [14]. Some researchers suggest that targeted parents contribute to their own alienation as a result; however, Haines et al. [5] argue that targeted parents may withdraw from their child after realising they cannot meet the alienating parent’s demands and after recognising when all avenues for resolution are exhausted. Targeted parents also discuss not wanting to disrupt or subject their children to conflict [5].

### 1.3. The Alienated Child

There is no evidence that specific characteristics or protective factors in children will increase or protect them from the likelihood of parental alienation occurring [5]. Despite this, Kelly and Johnston [13] suggest anxious, fearful, or overly passive children may lack the resilience to withstand the alienating process; however, the psychological consequences for children subjected to parental alienating behaviours are clear, with both negative immediate and long-term effects. These include self-esteem issues, anxiety, depression, substance use, increased suicidality, school-related difficulties, and a greater risk of being alienated from their children in the future [11]. Children exposed to parental alienating behaviours may develop a confused sense of self-perception and fail to remember how to trust their perceptions and feelings, resulting in an uncertain identity, lack of self-esteem, and deep insecurity [10,15]. These difficulties can lead to the inadequate and age-inappropriate development of independence and individuality. This can lead to an increased vulnerability to mental health disorders such as depression, anxiety, eating and feeding disorders, posttraumatic stress disorder (PTSD), and other psychosomatic disorders [15]. These difficulties can persist even when alienated children reunite with the targeted parent [16]. Given the devastating impact parental alienating behaviours can have on the alienated child and their future self, it is important to gain more insight into finding ways to resolve parental alienation, alienating behaviours, and their consequences [5]. Adults who experienced parental alienating behaviours in childhood may be able to provide a rich narrative and offer a deeper understanding of onset, maintaining factors, and consequences.

### 1.4. Previous Research

More research is needed exploring the experience of adults exposed to parental alienating behaviours during childhood [16]. Baker attempted to address this gap by conducting a qualitative study of 38 participants’ experiences of being exposed to parental alienating behaviours in childhood. Resulting themes of depression, poor self-esteem, feelings of guilt and shame, increased alcohol and substance abuse, and an increased likelihood of experiencing alienation from their children were found [11,16]. Baker’s studies, which provided a rich and detailed account of alienated adult children’s experiences, were nevertheless limited by offering a descriptive lens from a North American perspective [11,17].

Italian researchers Verrochio and colleagues [18] found that individuals who were exposed to parental alienating behaviours as children had a higher likelihood of developing low self-esteem, perceiving only negative aspects of situations, or having poorer coping skills in stressful environments [18]. In 2018, Verrocchio and colleagues [19] also found an association between depression in adults and reported exposure to parental alienating behaviours in childhood. Verrochio’s quantitative research offers a contained view of the effects of parental alienating behaviours on alienated adults’ health and wellbeing but does not completely capture the nature of the individual experiences of each participant.

Research by Bentley and Matthewson [20] attempted to address gaps in the literature by using an evidence-based thematic approach with an international sample of adults exposed to parental alienating behaviours in childhood (*n* = 10) to understand their lived experience. Bentley and Matthewson’s study was rich and vivid in detail but was limited by a small, gender-biased (eight female, two male) sample [20].

The present study aimed to add to the research by Bentley and Matthewson [20] by continuing to qualitatively investigate the lived experience of adults who were exposed to parental alienating behaviours in childhood. This study explored the association between exposure to parental alienating behaviours in childhood and mental health outcomes in adults. This study was exploratory. As such, hypothesis testing was not conducted and causal relationships between variables could not be established.

## 2. Method and Materials

### 2.1. Participants

A total of 20 participants engaged in semi-structured interviews (60–90 min) based on their experience of being exposed to parental alienating behaviours. Recruitment was open to people of all genders (male, female, transgender, non-binary, other, prefer not to say) 18 years and above who were exposed to parental alienating behaviours in childhood or adolescence. Prospective participants were recruited internationally through social media platforms such as Facebook and Instagram, and online parental alienation support groups. Prospective participants were screened to determine eligibility using the Baker Strategy Questionnaire, a 20-item measure designed to rate the frequency of exposure to parental alienation behaviours with responses ranging from 0 (never) to 4 (always) [21]. The BSQ has good internal consistency- α = 0.93 [21]. Scores range from 0–80, with higher scores indicating greater exposure to parental alienating behaviours [21]. All included participants had BSQ scores over 40. All included participants had endorsed item 2 on the BSQ “*Limited or interfered with my contact with the other parent such that I spent less time with him or her than I was supposed to or could have (limited contact)*”. Participants were advised about the nature and aim of the study before their semi-structured interview.

Socio-demographic data were collected to give a clearer indication of participants’ lives as adults. The data included current age, place of birth, education, relationship, employment status, and information about their experience of being exposed to parental alienating behaviours. Participants were aged between 26 and 59 (*M* = 41.05, *SD* = 10.19) with a gender ratio of 60% female and 40% male participants. The alienating parent was identified as the mother in 75% of cases and the father in 25% of cases. The mean age of separation from the targeted parent was 6.9 years of age (*SD* = 4.35 years of age). The majority (80%) of participants indicated they had since reunified with their targeted parent, 15% had somewhat reunified, and 1 participant had not reunified with their targeted parent. Table 1 provides a summary of the sample.

### 2.2. Procedure

The study was approved by the Tasmanian Social Sciences Human Research Ethics Committee (ethics reference number H0016616). Qualitative data were collected in semi-structured interviews of 60–90 min. Interviews were held online via Zoom or Skype. Once the interview commenced, participants were reminded of the limits to confidentiality and their right to withdraw from the study at any time. Participants were then invited to discuss their individual experiences of parental alienation and parental alienating behaviours using questions stems as prompts (question stems are available from the corresponding author). Using question stems as verbal cues allowed participants to elaborate on their unique experiences. At the end of the interview, participants were debriefed and given the opportunity to ask questions. The audio-recorded interviews were then transcribed verbatim, with each transcript representing an individual raw data point. Participants were then invited to amend and edit their transcripts for accuracy if they wished. Transcripts were then added to the raw dataset collected by Bentley and Matthewson [20] to create a larger raw dataset of 20 participants. All resulting raw data (*n* = 20) were coded and analysed together, according to the principles of thematic analysis using NVivo11 software.

### 2.3. Data Analysis

The approach to research, methodology, and data analysis was based on the theoretical paradigm of pragmatism [22]. Thematic analysis was chosen to examine participants’ experiences of parental alienation and exposure to parental alienating behaviours. Braun and Clarke’s thematic analysis was used to analyse the qualitative data [23]. Additionally, Forero and colleagues’ 4-dimension criteria were used to ensure the reliability, objectivity, and validity of the analysis [24]. Table 2 illustrates the stringent protocol applied in this study.

The datasets were analysed and coded by three independent researchers to minimise bias, with each participant transcript representing a unit of thematic analysis. Codes were developed based on the interview transcript of each participant and subsequently combined into themes. NVivo software was used to assist in the organisation and extraction of codes and themes.

## 3. Results

The participants described their perception of how exposure to parental alienating behaviours in their childhood affected their mental health in adulthood. There were 459 references to mental health impacts across the entire dataset. Four broad themes were identified: (1) Mental Health Difficulties; (2) Addiction and Substance Use; (3) Emotional Pain; (4) Coping and Resilience. Each theme is presented below using data extracts from individual participants, which represent their experience. Example quotes from participants’ transcripts are provided. These are their exact words. Larger themes have been broken down into sub-categories as described below.

### 3.1. Mental Health Difficulties

Participants reported experiencing a range of mental health difficulties. Some of these difficulties were self-diagnosed and some were reportedly formally diagnosed by a clinician. Participants also described their perception of how exposure to parental alienating behaviours impacted their adult lives. Although we cannot infer causation, it was the view of the participants that their exposure to parental alienating behaviours and experience of being alienated from a parent were associated with their mental health difficulties. There were 66 references to mental health difficulties across most of the dataset, with eight sub-themes identified and described below.

Depression and Anxiety: 55% of participants spoke about experiencing depression and anxiety in their adulthood and the impact they said it had on their ability to function in their daily lives:


*“I was a functioning depressed person where I would go to work, I could handle kids, but I would fall apart after that. And the psychiatrist said, “I’m prescribing you this” and he writes it down on his little tablet and he hands it to me, and it says, “move out of your (alienating parent) mother’s house.”*


Some participants wondered how their early experiences impacted their depression and anxiety:


*“…. It’s quite scary the depths in terms of my negativity and capability to go into depression…I don’t know to what extent this comes from having been denied the attention of a primary caregiver (alienating parent) for some of my formative years.”*


2.Eating Disorders/Body Image Issues: 20% of participants reported experiencing eating disorders and/or body image concerns that developed in adolescence. There was variability in accounts, with some participants able to explain how the origins of their difficulties arose compared to others who were less certain. The extent to which these difficulties persisted into adulthood for these participants was unclear:


*“I started getting an eating disorder, I had bulimia… I didn’t understand either, I didn’t understand why I was doing these things either…”*


3.Personality Difficulties: 40% of participants described difficulties related to personality dysfunction ranging from a formally diagnosed case of borderline personality disorder, (BPD) to a variety of difficulties, including emotion dysregulation, fear of abandonment, splitting, excessive reassurance and validation seeking, mistrust in self, impulsivity, inability to resist urges, and the need to impress others:


*“I have noticed as well is I have a very needy vibe in relationships where I’m capable of if I’m getting everything I need, possibly in a borderline narcissistic way…”*


Some participants also reported experiencing longstanding controlling or perfectionist tendencies which they related to exposure to parental alienating behaviours, mainly aimed at pleasing the alienating parent:


*“I’ve had… you know…really struggled with perfectionism because I didn’t really know what would set my parents off and if I wasn’t perfect, I’d get disciplined.”*


4.Posttraumatic Stress Disorder (PTSD): One participant reported suffering from diagnosed PTSD after living an unstable lifestyle. This participant described moving from one place to another and being exposed to unsafe people after her alienating parent reportedly forced her to move out of the family home at 17 years of age. This participant said the effects of these experiences still caused her difficulty in adulthood:


*“I was suffering from posttraumatic stress disorder and all I was doing was hiding like a, a mollusc in a shell, away from the world... I didn’t understand what the posttraumatic stress was, I didn’t understand why I was in a perpetual state of anxiety, and I couldn’t switch it off. It’s taken me 30 years to be able to understand that.”*


5.Psychosomatic Symptoms: 10% of the sample reported experiencing psychosomatic symptoms such as fibromyalgia, chronic fatigue, hypersensitivity to sound and the environment, cognitive “fog”, and alopecia, which some suspected were due to their exposure to parental alienating behaviours:


*“I do have moments still today where I can fog over, I have foggy moments, so it has affected me, and I’ve had would you call it chronic fatigue for a lot of life... but it had yeah affected me in a big way physically you know... people say it’s from abuse and stuff like that… I think it has a lot to do with what I’ve experienced, the constant grief…”*


6.Attention-Deficit/Hyperactivity Disorder (ADHD): One participant spoke about ADHD symptoms and possible related diagnoses. They wondered if they had these diagnoses based on the difficulties they had. They also wondered if these symptoms were related to their exposure to parental alienating behaviours in childhood:


*“I think I’ve got serious ADHD; I think I might have mild Asperger’s, I’m not sure... I’m possibly even bipolar, these are all the most likely what ifs, I’ve not got a diagnosis... I do wonder though that considering the symptoms of ADHD and considering I’ve probably had it at an early age, I do wonder if this alienation thing exacerbated it…”*


7.Self-Harm: 15% of participants described experiencing self-harm through cutting with incidents starting from the age of 11 and continuing into early adulthood:


*“When I was cutting my legs, I was only 11 years old…I was quite ashamed of that… I had this metal ruler that my brother had given me, and I was doing that but, on my legs, you know, and it was sort of… these little things that started happening over time became my normal.”*



*“I would hurt myself a lot and one time I ended up in a psychiatric hospital.”*


8.Suicidal Ideation: 30% of participants reported experiencing suicidal ideation from adolescence into adulthood. Some were able to link their suicidal ideation directly to their exposure to parental alienating behaviours:


*“I often had suicidal thoughts. That was throughout my 20s. So, I wouldn’t really want to relive like my age from 20 to 25. On an emotional level, it was a horrible life at times. So, I think a lot of emotional instability, but without being able for me at that point, to link it to what I lived, in childhood.”*


Some participants described their suicidal ideation plans in detail:


*“I thought about suicide where I wanted- I’d drive my Mustang 140 miles per hour down these country roads- You know how easy would it be for me to run into that tree?... I remember standing in the kitchen right before this emancipation thing (from alienating parent) happened, and there was a knife on the counter, and I considered it.”*


Further, 50% of the total sample reported becoming targeted parents in adulthood. Of these participants, four reported experiencing suicidal ideation and described how thoughts of their children stopped them from dying by suicide:


*“I did get suicidal more than once- I didn’t go through with it. The thing that did stop me was thinking about my kids and wanting to see them again.”*


### 3.2. Addiction and Substance Use

A total of 41 references were made across 55% of the dataset concerning addiction and substance use impacting participants in adulthood with three sub-themes identified and described below.

Alcohol: 55% of participants reported their alcohol consumption started in early adolescence and peaked in their late teens–early 20s:


*“I started drinking at 16 but that was more as an escape from home life when I was drinking with my friends from work. Because they’d have parties on weekends and whatever, but you know, I wouldn’t drink much back then. It wasn’t until I was in my 20′s that I drank a lot more.”*


One participant described how the smell of alcohol reminded her of her alienating parent, thus acting as a deterrent from alcohol for her:


*“I can’t drink wine I’ve only ever had two glasses of wine in my life—mum was a really big wine drinker she was such a big wine drinker and I’d be her little butler when she was drinking- I’d go and get more wine, so I’d go empty out half her bottle of wine and fill it up with water and she was so drunk she didn’t know. But yeah, I don’t touch wine. I hate the smell of wine…”*


2.Drugs: 35% of participants described using cannabis daily, occasionally, or reported a previous cannabis addiction; 20% of participants reported recreational use of MDMA; 5% of participants reported dangerous use of methamphetamine and hallucinogens. Some of these participants reported relying on these drugs when going out and consuming high doses of their drugs of choice. Some of these participants recalled using substances to numb painful emotions or to feel normal:


*“Anything to get away and not feel- but try and feel at the same time. It was a weird position to be in.”*


3.Sex/Pornography: 10% of participants spoke about leading promiscuous and impulsive lifestyles to find connections with others:


*“Drugs, alcohol, sex. Just went on a complete spiral out of control. Taking things so that you don’t feel, but then almost like an addiction trying to attach yourself to people so that you can feel something at the same time. Really quite strange.”*


### 3.3. Emotional Pain

Participants described how their exposure to parental alienating behaviours in childhood was associated with emotional pain in adulthood. There was a total of 83 references to emotional pain across 95% of the dataset. The theme of emotional pain was broken into eight sub-themes described below.

Shame and Guilt: 45% of the participants recalled feeling guilt or shame about their experience. Some had come to understand that although they knew their experience was not their fault, they could not avoid feelings of guilt that lasted into adulthood. Some participants were confused by their guilt or were unable to explain why they felt guilty:


*“There’s a lot of blame and guilt, and it’s still there like sometimes I’m like “oh, maybe that was, maybe this is my fault, maybe I have done this, maybe this is wrong, maybe that was lies, maybe this isn’t...” You know there’s a lot of, it’s still a lot of confusion, within myself about what to believe and what not to believe…”*


2.Self-Esteem: 40% of participants reported how being exposed to parental alienating behaviours in childhood had impacted their self-esteem into adulthood, with some feeling worthless and unequal to others. One participant reflected on how low confidence and self-esteem was linked back to not having a voice in childhood:


*“For so long, it didn’t matter who I met or from what walk of life they were—it could’ve been a street sweeper, it could’ve been a barrister, I immediately thought that I was the lesser person…I didn’t place a lot of value on myself, and I realise now that I was very vulnerable to being mistreated or abused. I realise now that at some level, I accepted this behaviour and agreed that it was what I deserved…I was always fearful that I’d get ripped off because I didn’t have a voice. I guess that’s what we learned as children- we never had a voice.”*


3.Loneliness and Isolation: 30% of participants reflected on loneliness and isolation. Some chose to isolate themselves from the outside world. Others felt lonely or isolated, which they attributed to exposure to parental alienating behaviours:


*“Sometimes you can feel a bit lost and forgotten, especially if you’re in a situation where you’re a fairly highly functioning human, but you have all these things that are still the background, and they just, they just remain there. And not many people understand, and so that can be quite lonely…”*


4.Helplessness: 20% of participants reflected on their feelings of helplessness in terms of their exposure to parental alienating behaviours in childhood or their current experience of being a targeted parent:


*“My God I’m still stuck in this mess that someone else created...”*


5.Grief and Loss: 60% of participants described their feelings of grief and loss with the most predominant responses involving a sense of loss around childhood, family, and denial of access to the targeted parent:


*“I’ve spent probably the last year, almost straight really grieving and mourning, having to work through this because it was so well hidden, it was so normalised, I was grieving. Grieving my childhood. Grieving the parents, I didn’t get. Grieving the person that I thought I was and who I actually was.”*


6.Anger: 45% of the participants reported feeling varying degrees of anger mainly aimed at the alienating parent. Some reported mild degrees of anger, while others were very specific in their resentment towards their alienating:


*“I blame my mother and fuck you, fuck you, you fucked it for a fucking long time, you fucked it love. And there’s a part of me that has such major resentment, major, you know…if she wasn’t so old, if I could drag her into court to sue for that, I would do it.”*


7.Abandonment: 15% of participants spoke about how their feelings of abandonment had impacted their lives as adults:


*“I don’t believe anyone’s going to stay.”*


8.Trust Issues: Other participants described how it was difficult for them to trust others, and themselves, due to their vulnerability:


*“At this point I find it sometimes hard with people, when I meet people to, to build up trust. I think parental alienation also, it causes a lot of.. trust issues.”*


### 3.4. Coping and Resilience

A total of 95% of participants made references to coping and resilience around their PA experience, with 269 references broken into three subthemes of maladaptive coping, adaptive coping, and meaning making.

Maladaptive Coping: 50% of participants reported using coping styles that were maladaptive, including stoicism, avoidance, indifference, mistrust, creating barriers, vengeful thinking against the alienating parent, and withdrawing, with 59 references made across the dataset.Adaptive Coping: participants described using adaptive coping strategies with a total of 56 references made by 80% of the group. They described using adaptive coping skills in adulthood to deal with their exposure to parental alienating behaviours in childhood. These strategies included cognitive reframing, acceptance, forgiveness, healing, self-education about PA, self-care, and engaging in therapy or support groups.Meaning Making: all participants spoke about trying to make meaning of their experience in adulthood with 154 references to topics such as using self-reflection and gaining perspective, confusion about their experience of being exposed to parental alienating behaviours in childhood; coming to the realisation the alienating parent was lying; memories; life lessons; conflicting thoughts about their experience and contributing to research into parental alienation.

## 4. Discussion

The current study aimed to continue to qualitatively investigate the lived experience of adults who were exposed to parental alienating behaviours in childhood. This study explored the association between exposure to parental alienating behaviours in childhood and mental health outcomes in childhood. The dataset (*n* = 20) was obtained from an international sample of adults who had been exposed to parental alienating behaviours in childhood. Those data were thematically analysed with the aim of extending the existing literature on parental alienation with an international perspective. This study provided insight into the mental health impact of exposure to parental alienating behaviours in childhood with the following four themes identified: mental health difficulties, substance use, emotional pain, and coping and resilience.

### 4.1. Mental Health Difficulties

All participants reported that their mental health had been impacted by being exposed to parental alienating behaviours. Further, 90% reported having specific mental health difficulties. These findings correspond with and build upon previous literature showing an association between exposure to parental alienating behaviours and mental health difficulties in adulthood [5,11,14,20,25]. Participants reported experiencing mental health difficulties ranging from depression and anxiety, attention-deficit/hyperactivity disorder (ADHD), self-harm, personality issues, eating disorders, body image issues, somatic symptoms, posttraumatic stress disorder (PTSD), and suicidal ideation. Developmental psychologists Ainsworth and Bowlby provided insight into the importance of early childhood attachments and experiences in laying the foundation for optimal development from birth and across the lifespan [26]. According to Ainsworth, parents provide a secure base for infants from which to explore the world. The bond between infant and parent strengthens when the infant feels safe, has their needs met, and experiences a continually warm and nurturing relationship with the primary caregiver. As a result, the infant begins to develop positive emotional regulation and self-soothing skills [27]. Bowlby posed that to grow up “mentally healthy”, children needed to experience both a secure environment where the parent and child find enjoyment, in addition to having social, economic, and health security [27]. When children do not experience warm and nurturing environments during their early years, their internal working model of the world can be affected, resulting in unstable attachments, reduced resilience, and over-developed fear centres in the brain [26,28].

When parental alienating behaviours are present in the family environment, the child is at least exposed to suboptimal conditions and, at worst, is exposed to a range of abusive behaviours such as coercion, control, manipulation, and neglect [5]. If a child’s threat appraisal system becomes heightened and unmanageable and caregivers do not actively help to regulate their physiological arousal, the child becomes unable to categorise their experiences successfully. The child’s ability to respond flexibly to a perceived threat is damaged over time, resulting in hypervigilance, problems reading social cues, unpredictable levels of emotional reactivity, intense feelings of fear, reduced memory function, and learning difficulties [28]. Higher levels of emotional reactivity in early childhood are hypothesised to be a predictor of numerous mental health issues in adulthood, including depression, impulsivity, increased suicidal ideation, and criminal behaviour [29]. Furthermore, genetic predisposition and social learning within the family context also serve to increase the probability that children are more likely to inherit their parent’s mental health issues and patterns of behaviour [30,31]. Thus, the finding that 90% of participants experienced mental health difficulties in adulthood is consistent with the literature.

Another consideration is that children exposed to early maltreatment are believed to be more vulnerable to a complex form of posttraumatic stress (CPTSD) [32]. CPTSD can occur in adulthood when social–emotional needs are neglected in childhood. This neglect causes the brain to prioritise stress monitoring over higher functional cortisol development [33]. CPTSD, as it appears in the latest revision of the ICD-11, recognises the impact of repeated interpersonal trauma and disturbances in self-organisation in addition to re-experiencing, hypervigilance, and avoidance [33]. Participants’ reports of their mental health challenges, including emotion dysregulation and ongoing interpersonal difficulties, appear suggestive of CPTSD. CPTSD is considered a less stigmatising and more accurate diagnosis than borderline personality disorder (BPD) when individuals experience dysregulation and interpersonal difficulties and also have a history of repeated relational trauma [33].

### 4.2. Addiction and Substance Use 

The current study found that over half of the participants had reported having substance use issues. This finding is consistent with previous research [5,11,20]. In her study on alienated adult children, Baker found that adults who had been exposed to parental alienating behaviours in childhood had used alcohol and/or drugs at some stage in their lives, usually late adolescence to early adulthood, to “escape” or numb emotions [11]. Participants in the current study gave the same reasons for using substances. Consistent with previous research, participants in the current study said they used substances to reduce the impact of parental alienating behaviours on their lives, to avoid emotions, or to create “chaos” when life became too stable.

A small number of participants spoke about their experiences of sex and pornography addiction as adults. To the authors’ knowledge, there is no research to date exploring the association between exposure to parental alienating behaviours in childhood and sex and/or pornography addiction in adulthood. Emerging research suggests that sex addiction is associated with adverse childhood experiences and insecure attachment styles (specifically anxious and fearful-avoidant) [34]. The participants in the current study who reported experiencing sex and pornography addiction described having been exposed to adverse events in their childhood. They also attributed their sex addiction to having insecure or disorganised attachment to their alienating parent.

Psychological theories of addiction range from the neurobiological explanation of the pleasure–withdrawal cycle, the behavioural explanation of stimulus–response cycles, and the cognitive-dysregulation explanations of the loss of self-control [35]; however, vulnerability factors should be considered when examining the origins of addiction and substance use in adults exposed to parental alienating behaviours in childhood [36]. The current study illustrated that more than 50% of participants were previously or currently engaged in substance use or addictive behaviours to varying degrees. Further research on the association between exposure to parental alienating behaviours and addiction and substance use would be beneficial.

### 4.3. Emotional Pain 

The current study found that 95% of participants experienced some form of emotional pain that they attributed to being exposed to parental alienating behaviours. Eight sub-categories were identified: shame and guilt; self-esteem; loneliness and isolation; helplessness; anger; abandonment; trust issues; grief and loss. Grief and loss were the most frequently described experience (60%). These findings are consistent with previous literature demonstrating an association between emotional pain and exposure to parental alienating behaviours in childhood [5,11,16,17,20]. Participants in the current study described feeling invalidated, invisible, and unrecognised by greater society. Recent research conducted by Harman, Matthewson, and Baker [10] focused on the losses experienced by alienated children. Their research explained how alienated children exposed to parental alienating behaviours suffered a gradual “cascade of losses” including: loss of individual self; loss of childhood and innocence; loss of a “good enough” parent; loss of extended family; loss of community. These losses lead to disenfranchised grief, especially in relation to time lost with the targeted parent [10]. Participants in the current study had difficulty describing the origins of their grief. Harman and colleagues’ [10] explanation of ambiguous loss leading to disenfranchised grief may help to clarify the grief and loss experience of adults exposed to parental alienating behaviours in childhood. Some participants were informed about disenfranchised grief and loss during their interview for this study. The researchers observed that these participants reacted to this information with a profound sense of relief, stating that the information had validated their experience.

### 4.4. Coping and Resilience 

Participants described the ways they coped with being exposed to parental alienating behaviours in childhood. Although 50% of participants described using maladaptive coping styles such as vengeful thinking about the alienating parent or avoiding memories of the alienating parent, more participants spoke about using adaptive coping strategies. All participants expressed a desire for their experience to mean something. Participants wanted to raise awareness of parental alienation, help others learn to identify parental alienation, and/or advocate for children currently exposed to parental alienating behaviours so they can be better protected by family court systems, child protection, and the broader community. There is ample research investigating the impact of adverse childhood events on the development of children through to adulthood with higher exposure to adversity associated with poorer mental health outcomes [37]; however, despite having faced exposure to parental alienating behaviours and the adversity associated with it (abuse, sexual abuse, witnessing domestic violence, fear, etc.), some participants said their experience had given them strength.

Resilience theory stems from the study of children who had overcome adversity against all odds [38]. Studies into resilience factors disputed the findings that all children facing adversity were more likely to develop psychopathology in adulthood [38]. This finding aligns with the stories relayed by some participants in the current study. Despite the adversity they faced in childhood, some were now living fully functioning, happy lives with healthy relationships with others. This supports the notion that adverse childhood experiences, such as exposure to parental alienating behaviours, can be moderated by individual protective factors such as inner locus of control and increased intrinsic motivation to alter life circumstances [38]. While participants spoke in terms of wanting to make meaning of their experience and wanting to overcome their childhood adversity, these sentiments appeared to be associated with two factors: increased age and increased education about parental alienation. Older participants who had had time to reflect more deeply on their experience and participants who had educated themselves on parental alienation through reading books, taking courses, and engaging in online support appeared to have a greater understanding and acceptance of their experience.

### 4.5. Key Findings

Drug use: over half the sample reported past or current substance use. Some described depending on daily cannabis use to get through the day or needing to use recreational drugs when engaging in social occasions to have a “good time.” Baker [11] also found one-third of participants used alcohol or drugs as a coping mechanism.Confusion about their experience: many participants described feeling confused about their experience of being exposed to parental alienating behaviours, especially when trying to make meaning of the past. Participants described feeling confused about their identity, their perceptions of reality, and failure to trust their judgement. These findings align with previous research showing that adults exposed to parental alienating behaviours can experience a confused sense of self, which appears to be associated with difficulties in identity development and mental health concerns in adulthood [15]. The confusion encountered by adults exposed to parental alienating behaviours in childhood may be related to the ambiguous losses they experience [10]. Additionally, participants who had sought help from therapists unfamiliar with parental alienation were left feeling invalidated and more confused about their experience.Education about parental alienation increased coping: education about parental alienation appears to be associated with the use of adaptive coping skills. This finding may be explained by Foucauldian discourse analysis in which knowledge is hypothesised to increase power [39]. This certainly appeared true for participants who were actively engaged in raising parental alienation awareness, advocacy, self-education, and therapy. These participants seemed better able to describe and reflect on their experiences more so than participants who were not engaged in activities increasing knowledge of parental alienation.Intergenerational transmission of parental alienation: The current study gave further weight to findings of previous research showing an intergenerational pattern of parental alienation transmission [11,20]. Of the participants interviewed (*n* = 20), 50% were parents who described being alienated from their children because of parental alienating behaviours. Participants who were now targeted parents disclosed profound difficulties coping with the fight to maintain relationships with their children; communicating with the alienating parent; attending and financing the ongoing legal battle; trying to find ways to cope with their losses, whilst prioritising their mental health. Some participants had realised they were drawn to partners resembling their alienating parent in terms of similar personality traits and patterns of behaviour. Most of these participants had tried unsuccessfully to regain contact with their children, with one mother capturing the essence of this predicament in her comment “you’re damned if you do, and you’re damned if you don’t”.Concerning suicidal ideation rate: almost one-third of participants described having past or current suicidal ideation. Many of these participants had attempted suicide. Most were targeted parents who appeared particularly distressed by a continuing cycle of past exposure to parental alienating behaviours and current exposure as an adult. All cited their children as a protective factor in stopping them from dying by suicide, despite non-fatal suicide attempts.

### 4.6. Practice Implications

Given that all participants described mental health concerns that they considered a result of their exposure to parental alienating behaviours, mental health clinicians working with this population need adequate preparation and knowledge to address these difficulties [40]. As previous research and the current study have illustrated, mental health clinicians are often under-equipped to provide therapy for this group [41]. Clinicians must reflect on their competencies when working with people exposed to parental alienating behaviours, because a lack of knowledge can result in further damage and invalidation.

We recommend that mental health clinicians have a thorough knowledge of attachment and relational disorders because this is likely to be a key area of work and necessary in helping people exposed to parental alienating behaviours rebuild and address their internal working models of relational attachment [26]. Additionally, clinicians should use a trauma-informed approach and be able to provide a secure base to people exposed to parental alienating behaviours in childhood given the likelihood of poor relational or attachment histories. Schema or Narrative Therapy may help people exposed to parental alienating to piece together and make sense of their experience. Understanding their stories will also help people exposed to parental alienating detect and address patterns of intergenerational transmission of parental alienation. Harman and colleagues’ [10] research on ambiguous loss is also likely to help clarify why people exposed to parental alienating might be confused about their experience. Finally, clinicians must undertake an ongoing thorough risk assessment and safety planning with their clients. Teaching coping skills such as distress tolerance, mindfulness, and acceptance, in combination with problem solving around potential reunification plans and assertive communication, may also help instil a sense of hope. Additionally, engaging in support groups may be useful in helping people exposed to parental alienating behaviours to increase their knowledge and understanding of parental alienation. Communicating with others in a similar situation may help to reduce feelings of isolation.

The current study revealed the importance of intervening and supporting children and families exposed to parental alienating behaviours because many participants said they wished they had received support earlier. School psychologists and counsellors, equipped with adequate knowledge of parental alienation and parental alienating behaviours, are in a good position to begin early intervention work with children and families. Similarly, conducting educational professional development workshops focusing on parental alienation and parental alienating behaviours will enable teachers and support staff to be more aware of the issues involved.

## 5. Limitations and Future Directions

All efforts were made to conduct a thorough analysis and interpretation of data; however, there were limitations to the current study. Firstly, quantitative data rely on the account of the individual being interviewed to be a true interpretation of the facts [23]. Participants’ accounts given in the current study were accepted as factual despite being subjected to self-report bias. Furthermore, while the Baker Strategy Questionnaire was used as a screening tool to determine if participants had been exposed to parental alienating behaviours in childhood, it is not clear if all participants were accurately able to distinguish their experience from high conflict divorce or custody disputes. Secondly, results are not generalisable as participants only represented countries from developed nations despite efforts to recruit a globally representative sample. Due to cultural, social, and economic differences, cross-cultural considerations in the experience of exposure to parental alienating behaviours need further investigation.

Recruitment of participants was opportunistic and may not be representative of the entire population of people exposed to parental alienating in childhood. Some participants stated that they had been formally diagnosed with specific mental health conditions or they opined that they might have specific mental health disorders. Without conducting our own formal, valid diagnostic assessment, we cannot confirm the presence of these conditions in the current sample. Further research needs to examine if adults exposed to parental alienating behaviours in childhood experience particular mental health conditions more than people who have not been exposed to parental alienating behaviours. If it is accepted that parental alienating behaviours are a form of child abuse [5], people exposed to parental alienating behaviours may likely develop symptoms of complex posttraumatic stress disorder (CPTSD). Further research needs to establish if there is a direct relationship between exposure to parental alienating behaviours and later diagnosis such as (CPTSD). Future research should build on the rich, qualitative data captured in this study by using a variety of methodological approaches. Considering the lack of awareness of parental alienation and alienating behaviours as a broader, systemic social issue, future research must endeavour to come to a unified definition of parental alienation and parental alienating behaviours with the aim of making these easily identifiable in the wider community.

## 6. Conclusions

This study highlights the pervasive impact of exposure to parental alienating behaviours in childhood on the mental health of adults. Key findings illustrate the need for validating and research-informed interventions for children, adolescents, and adults exposed to parental alienating behaviours. Parental alienation awareness campaigns and engagement in support networks may be beneficial.


*“Let’s hope this is the beginning of it… this is the beginning of us, as a society starting to realise… that no matter what happens between parents, that children are not weapons that can be used against the other.”*
(From an adult alienated child participant)

## Figures and Tables

**Table 1 children-09-00475-t001:** Demographic data alienated adult children study.

ID	Age	Gender	Country of Birth	Current Location	Marital Status	Educational Level	Employment Status	Alienated from	Age of Separation	Reunification Status
1.	37	M	Chile	Australia	Separated	TAFE	Employed	Father	3	Yes
2.	30	F	USA	Australia	De Facto	Undergrad	Employed	Father	13	Yes
3.	35	F	Australia	Australia	De Facto	Undergrad	Employed	Father	2	Somewhat
4.	53	M	Australia	Australia	Maried	Undergrad	Employed	Father	5	Somewhat
5.	52	F	USA	USA	Divorced	Masters	Employed	Father	4	Yes
6.	49	F	Netherlands	Netherlands	Single	Undergrad	Employed	Father	12	Yes
7.	50	M	Australia	Australia	Widow	High School	Unemployed	Father	2	Yes
8.	35	M	Australia	Australia	De Facto	Diploma	Unemployed	Father	7	Yes
9.	30	F	Thailand	Australia	De Facto	TAFE	Employed	Father	1	Yes
10.	59	F	USA	USA	Divorced	PhD	Unemployed	Father	13	No
11.	47	F	Australia	Australia	Divorced	Undergrad	Employed	Father	2	Yes
12.	54	F	Australia	Australia	Single	High School	Volunteer	Father	3	Yes
13.	35	F	Englanda	Thailand	Single	Undergrad	Employed	Mother	8	Yes
14.	44	F	Australia	Australia	Partner	Diploma	Employed	Mother	10	Yes
15.	49	F	Australia	Australia	Married	Undergrad	Employed	Father	7	Somewhat
16.	26	F	USA	Australia	Single	High School	Employed	Mother	13	Yes
17.	45	F	Belgium	Belgium	Married	Masters	Employed	Father	7	Yes
18.	30	M	England	Australia	Single	High School	Employed	Father	11	Yes
19.	33	F	England	Australia	Single	Honours	Student	Mother	3	Yes
20.	28	F	Germany	Germany	Partner	Masters	Student	Mother	12	Yes

**Table 2 children-09-00475-t002:** Study protocol.

Criteria	Purpose	Strategy	Additional Rigour
Credibility	To establish confidence that participants’ accounts were true/credible	To recruit from support groups To use tools to screen for exposure to parental alienating behaviours	Investigators spent time liaising with potential participants before the study to ensure suitability Investigators had a good theoretical understanding of parental alienation before beginning interviews Regular debriefings were held with additional members of the lab team to discuss issues related to data collection
Dependability	To ensure findings made in the current study are repeatable	NVivo used to capture data analysis Detailed description of the method	A detailed track record of the data collection was kept outlining each step of the process Each stage of analysis was logged using NVivo in a stepwise fashion for ease of reference
Confirmability	To establish confidence that similar results would be confirmed by other researchers	To reflect on own biases and assumptions Triangulation of data	Frequent reflective supervision/correspondence with head investigator Data were triangulated from principles/theory, methodology, and interviews
Transferability	The degree to which results can be transferred to other settings	Data saturation	Data saturation was achieved when no new information was able to be obtained from transcripts after multiple coding sessions from 2 investigators

## Data Availability

The dataset can be obtained from the corresponding author upon request.

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
