# Peer review of "The Impact of Parental Alienating Behaviours on the Mental Health of Adults Alienated in Childhood"

_children, 2022, doi:10.3390/children9040475_

Round 1

Reviewer 1 Report

This is an interesting and important paper.  Parental alienation is a serious mental condition affecting millions of children, and it is important to realize that it has consequences that continue into adulthood.  (In fact, the mental consequences in adulthood appear to persist even when the individuals have reunited with their previously rejected parents; the authors may want to point that out.)

My main concern is that the authors do not adequately distinguish between parental alienation (the mental condition of the alienated child) and alienating behaviors (the activities of the favored parent that cause parental alienation in the child).  They should clearly explain that distinction at the outset and then use it consistently through the paper.  Otherwise, the following confusions occurs ...

Section 2.1:  The authors explain how they determined the participants were exposed to alienating behaviors (through the BSQ), but they do not explicitly explain how they knew the participants actually became alienated from the rejected parents.  Alienating behaviors are very common in divorced families, but it is relatively rare for the child to actually become alienated.  (The relative prevalences are perhaps 10:1.)  

Page 3, line 105.  They say, "who believed they were exposure [sic] to PA as children."  They actually mean that the subjects in the Verrochio study were exposed to alienating behaviors, not PA.

Page 5, line 180.  "Exposure to parental alienation behaviour" should be "exposure to parental alienating behaviour."

Page 5, line 190.  "Exposure to PA in childhood" probably should be "exposure to parental alienating behaviour in childhood."  However, it could be either PA or ABs, depending on the exact meaning of the sentence.  This confusion will be avoided if the authors consistently use PA for the experience of the child and ABs for the activities of the favored parent.

Page 11, line 518.  "AAC exposed to PA in childhood."  This is unclear.  Do the authors mean "AAC exposed to alienating behaviors in childhood" or "AAC who experienced parental alienation in childhood."

There are many more examples of this same ambiguity throughout the paper, which need to be corrected.

Page 6, several places.  "Posttraumatic" does not have a hyphen, in DSM usage.

Page 6, line 240.  ADHD is spelled:  attention-deficit/hyperactivity disorder. 

Author Response

We have revised the introduction to provide definitions that better distinguish between parental alienation and alienating behaviours. We have revised the entire manuscript to avoid confusion between parental alienation and parental alienating behaviours. We have updated the spelling of “posttraumatic" and “attention-deficit/hyperactivity disorder.” We have included additional information about how the BSQ was used to determine participation inclusion and the presence of parental alienation.

Reviewer 2 Report

The article is interesting as it addresses a current issue of parental alienation and its impact on mental health.

Abstract:

The abstract correctly addresses the sections of the article.

Introduction:

It is complete and with updated references, in addition to exposing the limitations found in other articles and the purpose of the present one as a complement to the previous ones.

Method and Materials

This section is clear.

Results:

It is suggested that the authors design a table with the description of the results found in each of the four general themes found; since it would be easier to read and analyze the results found. Each of the analyzed themes has sub-themes that turn out to be very interesting; however, it would be clearer to include some tables with the information found and its impact on the variables studied.

In row 184 the number 3 of the theme found is repeated, it should be changed to number 4 coping and resilience.

In row 224 the following abbreviation PTSD is expressed but it does not describe the term or variable.

It is recommended to check that all abbreviated variables have the variable description in all sections of the article.

Discussion.

This section was presented by the authors in a clear and precise manner, with very interesting contributions and limitations.

It was very interesting to read the whole document, many congratulations for the work that was interesting to read and to know the contributions of the authors to this relevant topic in the area of mental health.

Author Response

Thank you for the positive feedback and congratulations. We have included a table in the results sections summarising the four general themes found. This is Table 3. We have corrected the number of the coping and resilience theme. We have stated all terms before giving the acronym and we have done so in each section of the manuscript.

Reviewer 3 Report

The paper is a qualitative description of the parental alienation and mental health symptoms reported by 20 participants. The authors used a qualitative methodology. Therefore, they described a codification from the interviews throughout the review of three independent researchers. Thus, the authors concluded that mental health disorders or symptoms mainly resulted from the alienation experienced in childhood or adolescence, not just for the participants but in the general population. Nevertheless, some concerns arose to be addressed by the authors. First, they considered the formal mental health diagnosis and informal mental health symptoms as equals. In other words, they assumed participants´ descriptions about their feelings as an indeed suffered disorders in them.
Moreover, the methodology used in the study doesn´t allow the authors to conclude a causal relationship between the variables of interest: parental alienation and mental health. Thus, a qualitative study with 20 participants, all according to the BSQ suffering from parental alienation, without a contrast group, where the condition of interest was absent, doesn´t allow the authors to assure how different could be the prevalence of mental health symptoms or disorders as a result of the parental alienation. It means, in general, the mental health symptoms can be present in the population without parental alienation.
Consequently, the general recommendations are:
1) to review the language and grammar all around the paper. 
2) to declare that the exploratory dimension of the study allows only considering a relationship of associative character between variables but not one of causal type. 
3) to correct their conclusions avoiding the stigmatization of the participants. 
4) to convert the assumption that participants were suffering from any mental health disorders without a formal, valid diagnostic, or an accurate applied tool or strategy, avoiding the causal inference of a relationship between parental alienation and mental health disorders or symptoms. 
4) to declare the necessity that future research should address how mental health symptoms grow due to parental alienation over the already expected prevalence of mental health as a consequence of all other social determinants in life.

Author Response

Language and grammar have been revised throughout the manuscript.  We did not change grammatical errors in participant statements because we did not want to deviate from their own words. Statements have been made in the manuscript about the exploratory nature of the study in the introduction, method, and discussion. Causal language has been changed to discuss associations between variables, unless the participants themselves inferred a causal relationship. In these instances, we have highlighted that this is the view of the participants.  We have taken great care to treat the voices of participants with respect and to do justice to their experience when drawing conclusions from the information they provided in the study. We have reviewed the language used in our conclusions again to ensure we have not said anything that is stigmatising. We have also had out manuscript reviewed by the founding director of EMMM. She is of the view that our manuscript is validating and in no way stigmatising.

We have updated the limitations sections of the manuscript. We have made it clear that it was participants themselves who reported to us the mental health conditions they have been diagnosed with or they believe they have. These are not our conclusions. We have highlighted in the discussion the need for research to further explore if people exposed to parental alienating behaviours are likely to experience particular mental health disorders.

Reviewer 4 Report

Review the grammatical mistakes, and few statistical ones. 

The grammatical mistakes are  in yellow and the one I have corrected are underlined. The statistical mistake I was referring to was the paragraph "Drugs: participants reported using a range of drugs in varying levels and for various..."   because it is not stated how many of the participants have used drugs 

Author Response

All grammatical errors highlighted in the body of the manuscript have been attended to. We have not changed the grammatical errors highlighted in the quotes from participants. This is because these are the words of participants verbatim. We do not want to alter their words. We have added a sentence to that effect at the start of the results section of the manuscript. Further details on participants’ drug use have been included.

Round 2

Reviewer 1 Report

Line 34.  The authors used Baker (2020) as the citation for the Five-Factor Model.  That citation is correct, but a better or additional citation is Bernet, 2022, The Five Factor Model for the Diagnosis of Parental Alienation. (file is attached).  It is a stronger citation because it is in a peer reviewed journal, rather than a book chapter.

Line 92.  There is a citation to reference 19, which is Clawar and Rivlin (1991).  It would probably be better to cite the second edition of that same book, which is Children Held Hostage: Identifying Brainwashed Children, Presenting a Case, and Crafting Solutions (2013).

Line 117.  Johnstone should be Johnston.

Line 203-205.  "Baker's studies ... perspective" is a run-on sentence.  A possible redo would be:  "Baker's studies, which provided a rich and detailed account of alienated adult children's experiences, were nevertheless limited by offering a descriptive lens from a North American perspective."

Table 2.  It is hard to read because each line is centered within the column.  It would probably be better to justify left each column and use bullets that are "hanging."

Line 764.  "That data" should be "Those data."

Line 1024.  "alienated adult child" should be "alienated adult children."

Line 1761.  "alienating parent in term" should be "alienating parent in terms."

Line 2012.  The quotation under Conclusion probably should have a citation.

Author Response

Thank you for the feedback. Our responses are in bold. 

Line 34.  The authors used Baker (2020) as the citation for the Five-Factor Model.  That citation is correct, but a better or additional citation is Bernet, 2022, The Five Factor Model for the Diagnosis of Parental Alienation. (file is attached).  It is a stronger citation because it is in a peer reviewed journal, rather than a book chapter.

Thank you for a copy of this article. We have changed this citation. 

Line 92.  There is a citation to reference 19, which is Clawar and Rivlin (1991).  It would probably be better to cite the second edition of that same book, which is Children Held Hostage: Identifying Brainwashed Children, Presenting a Case, and Crafting Solutions (2013).

We updated this citation.

Line 117.  Johnstone should be Johnston.

This is corrected.

Line 203-205.  "Baker's studies ... perspective" is a run-on sentence.  A possible redo would be:  "Baker's studies, which provided a rich and detailed account of alienated adult children's experiences, were nevertheless limited by offering a descriptive lens from a North American perspective."

We have used the suggested redo. 

Table 2.  It is hard to read because each line is centered within the column.  It would probably be better to justify left each column and use bullets that are "hanging."

We have reformatted the table. 

Line 764.  "That data" should be "Those data."

This is corrected. 

Line 1024.  "alienated adult child" should be "alienated adult children."

This is corrected. 

Line 1761.  "alienating parent in term" should be "alienating parent in terms."

This is corrected. 

Line 2012.  The quotation under Conclusion probably should have a citation.

We clarified this is a participant quote.